

# OZO v.1.0: Software for solving a generalized omega equation and the Zwack-Okossi height tendency equation using WRF model output

Mika Rantanen[1], Jouni Räisänen[1], Juha Lento[2], Oleg Stepanyuk[1], Olle Räty[1], Victoria A. Sinclair[1], and Heikki Järvinen[1]

[1]Department of Physics, University Of Helsinki, Finland
[2]CSC - IT Center for Science, Espoo, Finland

*Correspondence to:* Mika Rantanen (mika.p.rantanen@helsinki.fi)

**Abstract.** A software package (OZO, Omega-Zwack-Okossi) was developed to diagnose the processes that affect vertical motions and geopotential height tendencies in weather systems simulated by the Weather Research and Forecasting (WRF) model. First, this software solves a generalized omega equation to calculate the vertical motions associated with different physical forcings: vorticity advection, thermal advection, friction, diabatic heating, and an imbalance term between vorticity and temperature tendencies. After this, the corresponding height tendencies are calculated with the Zwack-Okossi tendency equation. The resulting height tendency components thus contain both the direct effect from the forcing itself and the indirect effects (related to the vertical motion induced by the same forcing) of each physical mechanism. This approach has an advantage compared with previous studies with the Zwack-Okossi equation, in which vertical motions were used as an independent forcing but were typically found to compensate the effects of other forcings.

The software is tailored to use input from WRF simulations with Cartesian geometry. As an illustration, results for an idealized 10-day baroclinic wave simulation are presented. An excellent agreement is found between OZO and the direct WRF output for both the vertical motion (correlation 0.97 in the midtroposphere) and the height tendency fields (correlation 0.95-0.98 in the whole troposphere). The individual vertical motion and height tendency components demonstrate the importance of both adiabatic and diabatic processes for the simulated cyclone. OZO is an open source tool for both research and education, and the distribution of the software will be supported by the authors.

## 1 Introduction

Today, high-resolution atmospheric reanalyses provide a three-dimensional view on the evolution of synoptic-scale weather systems (Dee et al., 2011; Rienecker et al., 2011). On the other hand, simulations by atmospheric models allow exploring the sensitivity of both real-world and idealized weather systems to factors such as the initial state (e.g., Leutbecher et al., 2002; Hoskins and Coutinho, 2005), lower boundary conditions (e.g., Elguindi et al., 2005; Hirata et al., 2016) and representation of sub-grid scale processes (e.g., Wernli et al., 2002; Liu et al., 2004; Beare, 2007). Nevertheless, the complexity of atmospheric dynamics often makes the physical interpretation of reanalysis data and model output far from simple. Therefore, there is also





a need for diagnostic methods that help to separate the effects of individual dynamical and physical processes on the structure
and evolution of weather systems.

Two variables that are of special interest in the study of synoptic-scale weather systems are the geopotential height tendency
and vertical motion (Holton and Hakim, 2012). Height tendencies are directly related to the movement and intensification or

decay of low and high pressure systems. Vertical motions affect atmospheric humidity, cloudiness and precipitation. They also
play a crucial role in atmospheric dynamics by inducing adiabatic temperature changes, by generating cyclonic or anticyclonic
vorticity, and by converting available potential energy to kinetic energy (Lorenz, 1955; Holton and Hakim, 2012).

For the need of diagnostic tools, some software packages have been developed to separate individual forcings contributions
to vertical motion and height tendency. DIONYSOS (Caron et al., 2006), a tool for analyzing numerically simulated weather

systems, provides currently online daily diagnostics for the output of numerical weather prediction models. RIP4 (Stoelinga,
2009) can calculate Q-vectors (Hoskins et al., 1978; Holton and Hakim, 2012) and a quasi-geostrophic (QG) vertical motion
(Holton and Hakim, 2012) from WRF model output, but the division of the $\omega$ to contributions of various atmospheric processes
is not possible in RIP4. Furthermore, many research groups have developed tools for their own needs, but do not have resources
to distribute the software.

Here, we introduce a software package (OZO) that can be used for diagnosing the contributions of different dynamical
and physical processes to atmospheric vertical motions and height tendencies. OZO calculates vertical motion from a quasi-
geostrophic and a generalized omega equation (Räisänen, 1995), while height tendencies are calculated using the Zwack-Okossi
tendency equation (Zwack and Okossi, 1986; Lupo and Smith, 1998). OZO has been tailored to use output from Weather
Forecast and Research (WRF) model (Wang et al., 2007; Shamarock et al., 2008) simulations run with idealized Cartesian

geometry. Due to the wide use of the WRF model, we expect OZO to be a useful open source tool for both research and
education.

In the following, we first introduce the equations solved by OZO: the two forms of the omega equation in Section 2.1 and the
Zwack-Okossi height tendency equation in Section 2.2. The numerical techniques used in solving these equations are described
in Section 3. We have tested the software using output from an idealized WRF simulation described in Section 4. Section 5

provides some computational aspects of the software. The next two sections give an overview of the vertical motion (Section
6) and height tendency (Section 7) calculations for this simulation. The conclusions are given in Section 8.

## 2   Equations

### 2.1   Omega equation

The omega equation is a diagnostic tool for estimating atmospheric vertical motions and studying their physical and dynamical

causes. Its well-known QG form, obtained by combining the QG vorticity and thermodynamic equations, infers vertical motion
from geostrophic advection of absolute vorticity and temperature (Holton, 1992):

$$L_{QG}(\omega) = F_{V(QG)} + F_{T(QG)} \tag{1}$$



where

$$L_{QG}(\omega) = \sigma_0(p)\nabla^2\omega + f^2\frac{\partial^2\omega}{\partial p^2} \tag{2}$$

and the two right-hand-side (RHS) terms are

$$F_{V(QG)} = f\frac{\partial}{\partial p}\left[\boldsymbol{V_g}\cdot\nabla(\zeta_g + f)\right] \tag{3}$$

$$F_{T(QG)} = \frac{R}{p}\nabla^2\left(\boldsymbol{V_g}\cdot\nabla T\right) \tag{4}$$

(the notation is conventional, see Table 1 for an explanation of the symbols).

Qualitatively, the QG omega equation indicates that cyclonic (anticyclonic) vorticity advection increasing with height and warm (cold) advection should induce rising (sinking) motion in the atmosphere. However, when deriving this equation,
ageostrophic winds, diabatic heating and friction are neglected. In addition, hydrostatic stability is treated as a constant and several terms in the vorticity equation are omitted. Although Eq. (1) often provides a reasonable estimate of synoptic-scale vertical motions at extratropical latitudes (Räisänen, 1995), these approximations inevitably deteriorate the accuracy of the QG omega equation solution.

The omega equation can be generalized by relaxing the QG approximations (e.g., Krishnamurti, 1968; Pauley and Nieman,
1992; Räisänen, 1995). Here we use the formulation

$$L(\omega) = F_V + F_T + F_V + F_Q + F_A \tag{5}$$

where

$$L(\omega) = \nabla^2(\sigma\omega) + f(\zeta + f)\frac{\partial^2\omega}{\partial p^2} - f\frac{\partial^2\zeta}{\partial p^2}\omega + f\frac{\partial}{\partial p}\left[\boldsymbol{k}\cdot\left(\frac{\partial\boldsymbol{V}}{\partial p}\times\nabla\omega\right)\right] \tag{6}$$

is a generalized form of (2) and the RHS terms have the expressions

$$F_V = f\frac{\partial}{\partial p}\left[\boldsymbol{V}\cdot\nabla(\zeta + f)\right], \tag{7}$$

$$F_T = \frac{R}{p}\nabla^2\left(\boldsymbol{V}\cdot\nabla T\right), \tag{8}$$

$$F_F = -f\frac{\partial}{\partial p}\left[\boldsymbol{k}\cdot(\nabla\times\boldsymbol{F})\right], \tag{9}$$


$$F_Q = -\frac{R}{c_p p}\nabla^2 Q, \tag{10}$$



$$F_A = f\frac{\partial}{\partial p}\left(\frac{\partial \zeta}{\partial t}\right) + \frac{R}{p}\nabla^2\left(\frac{\partial T}{\partial t}\right), \tag{11}$$

Apart from the reorganization of the terms in Eq. (6), this generalized omega equation is identical with the one used by Räisänen (1995). It follows directly from the isobaric primitive equations, which assume hydrostatic balance but omit the other

approximations in the QG theory. The first four terms on the RHS represent the effects of vorticity advection ($F_V$), thermal advection ($F_T$), friction ($F_F$) and diabating heating ($F_Q$). The last term ($F_A$) describes imbalance between the temperature and vorticity tendencies. For constant $f$ and constant $R$, the imbalance term is directly proportional to the pressure derivative of the ageostrophic vorticity tendency (Räisänen, 1995).

Because the operators $L_{QG}$ and $L$ are linear, the contributions of the various RHS terms to $\omega$ can be calculated separately if

homogeneous boundary conditions ($\omega = 0$ at horizontal and vertical boundaries) are used. These contributions will be referred to as $\omega_X$, where $X$ identifies the corresponding forcing term.

## 2.2 Zwack-Okossi height tendency equation

In the Zwack-Okossi method (Zwack and Okossi, 1986; Lupo and Smith, 1998), height tendencies are calculated from the geostrophic vorticity tendency. Neglecting the variation of the Coriolis parameter,

$$\zeta_g = \frac{g}{f}\nabla^2 Z \tag{12}$$

and hence

$$\frac{\partial Z}{\partial t} = \nabla^{-2}\left(\frac{f}{g}\frac{\partial \zeta_g}{\partial t}\right). \tag{13}$$

The geostrophic vorticity tendency at level $p_L$ is obtained from the equation

$$\frac{\partial \zeta_g}{\partial t}(p_L) = \frac{1}{p_s - p_t}\left[\int_{p_t}^{p_s}\left(\frac{\partial \zeta}{\partial t} - \frac{\partial \zeta_{ag}}{\partial t}\right)dp - \frac{R}{f}\int_{p_t}^{p_s}\left(\int_{p}^{p_L}\nabla^2\frac{\partial T}{\partial t}\frac{dp}{p}\right)dp\right] \tag{14}$$

where $p_s$ (here 1000 hPa) and $p_t$ (here 100 hPa) are the lower and the upper boundaries of the vertical domain. The vorticity tendency $\frac{\partial \zeta}{\partial t}$ is calculated from the vorticity equation (Eq. 2 in Räisänen (1997)) and the temperature tendency $\frac{\partial T}{\partial t}$ from the thermodynamic equation (Eq. 3.6 in Holton and Hakim (2012)). The ageostrophic vorticity tendency $\frac{\partial \zeta_{ag}}{\partial t}$ is estimated from time series of vorticity and geostrophic vorticity (Eq. 12) using centered time differences

$$\frac{\partial \zeta_{ag}}{\partial t} \approx \frac{\Delta(\zeta - \zeta_g)}{\Delta t}. \tag{15}$$

For the calculations shown in this paper, $\Delta t = 7200\ s$ was used.

In Eq. (14), the first integral gives the mass-weighted vertical average of the geostrophic vorticity tendency between the levels $p_s$ and $p_t$. The difference between the geostrophic vorticity tendency at level $p_L$ and this mass-weighted average is





obtained from the second double integral. In this latter integral, hydrostatic balance is assumed to link the pressure derivative of the geostrophic vorticity tendency to the Laplacian of the temperature tendency.

Analogously with the vertical motion, the height tendency is divided in OZO to the contributions of different physical and dynamical processes as

$$5 \quad \frac{\partial Z}{\partial t} = \left(\frac{\partial Z}{\partial t}\right)_V + \left(\frac{\partial Z}{\partial t}\right)_T + \left(\frac{\partial Z}{\partial t}\right)_F + \left(\frac{\partial Z}{\partial t}\right)_Q + \left(\frac{\partial Z}{\partial t}\right)_A . \tag{16}$$

By substituting the vorticity equation and the thermodynamic equation to Eq. (14), and then combining Eq. (13) and Eq. (14), the expressions for the RHS components of Eq. (16) are derived as follows:

(i) Vorticity advection (V) and friction (F) have a direct effect on the vorticity tendency in Eq. (14).

(ii) Thermal advection (T) and diabatic heating (Q) have a direct effect on the temperature tendency in Eq. (14).

10     (iii) The ageostrophic vorticity tendency in Eq. (14) is attributed to the imbalance term (A).

(iv) All five terms also affect the vorticity and temperature tendencies indirectly via vertical motions, which are calculated for each of them separately with the generalized omega equation.

This results in the following new expressions:

$$\left(\frac{\partial Z}{\partial t}\right)_V = \frac{f}{g(p_s - p_t)} \nabla^{-2} \left[ \int_{p_t}^{p_s} \left( -\boldsymbol{V} \cdot \boldsymbol{\nabla}(\zeta + f) - \omega_V \frac{\partial \zeta}{\partial p} + (\zeta + f)\frac{\partial \omega_V}{\partial p} + \boldsymbol{k} \cdot \left( \frac{\partial \boldsymbol{V}}{\partial p} \times \boldsymbol{\nabla}\omega_V \right) \right) dp \right.$$

$$\left. - \frac{R}{f} \int_{p_t}^{p_s} \left( \int_{p}^{p_L} \nabla^2 \left( S\omega_V \right) \frac{dp}{p} \right) dp \right] \tag{17}$$

$$\left(\frac{\partial Z}{\partial t}\right)_T = \frac{f}{g(p_s - p_t)} \nabla^{-2} \left[ \int_{p_t}^{p_s} \left( -\omega_T \frac{\partial \zeta}{\partial p} + (\zeta + f)\frac{\partial \omega_T}{\partial p} + \boldsymbol{k} \cdot \left( \frac{\partial \boldsymbol{V}}{\partial p} \times \boldsymbol{\nabla}\omega_T \right) \right) dp \right.$$

$$\left. - \frac{R}{f} \int_{p_t}^{p_s} \left( \int_{p}^{p_L} \nabla^2 \left( -\boldsymbol{V} \cdot \boldsymbol{\nabla}T + S\omega_T \right) \frac{dp}{p} \right) dp \right] \tag{18}$$

$$\left(\frac{\partial Z}{\partial t}\right)_F = \frac{f}{g(p_s - p_t)} \nabla^{-2} \left[ \int_{p_t}^{p_s} \left( \boldsymbol{k} \cdot \boldsymbol{\nabla} \times \boldsymbol{F} - \omega_F \frac{\partial \zeta}{\partial p} + (\zeta + f)\frac{\partial \omega_F}{\partial p} + \boldsymbol{k} \cdot \left( \frac{\partial \boldsymbol{V}}{\partial p} \times \boldsymbol{\nabla}\omega_F \right) \right) dp \right.$$

$$\left. - \frac{R}{f} \int_{p_t}^{p_s} \left( \int_{p}^{p_L} \nabla^2 \left( S\omega_F \right) \frac{dp}{p} \right) dp \right] \tag{19}$$

$$\left(\frac{\partial Z}{\partial t}\right)_Q = \frac{f}{g(p_s - p_t)} \nabla^{-2} \left[ \int_{p_t}^{p_s} \left( -\omega_Q \frac{\partial \zeta}{\partial p} + (\zeta + f)\frac{\partial \omega_Q}{\partial p} + \boldsymbol{k} \cdot \left( \frac{\partial \boldsymbol{V}}{\partial p} \times \boldsymbol{\nabla}\omega_Q \right) \right) dp \right.$$

$$20 \quad \left. - \frac{R}{f} \int_{p_t}^{p_s} \left( \int_{p}^{p_L} \nabla^2 \left( \frac{Q}{c_p} + S\omega_T \right) \frac{dp}{p} \right) dp \right] \tag{20}$$





$$\left(\frac{\partial Z}{\partial t}\right)_A = \frac{f}{g(p_s - p_t)}\nabla^{-2}\left[\int\limits_{p_t}^{p_s}\left(-\frac{\partial \zeta_{ag}}{\partial t} - \omega_A\frac{\partial \zeta}{\partial p} + (\zeta + f)\frac{\partial \omega_A}{\partial p} + \boldsymbol{k}\cdot\left(\frac{\partial \boldsymbol{V}}{\partial p}\times\boldsymbol{\nabla}\omega_A\right)\right)dp\right.$$
$$\left. -\frac{R}{f}\int\limits_{p_t}^{p_s}\left(\int\limits_{p}^{p_L}\nabla^2\left(S\omega_A\right)\frac{dp}{p}\right)dp\right] \tag{21}$$

The equation system used in this study has been adopted partly from Räisänen (1997). However, whereas Räisänen (1997) used the vorticity equation and the nonlinear balance equation to obtain height tendencies, the Zwack-Okossi equation is used

here. The main advantage of this choice is its smaller sensitivity to numerical errors. Our method produces quite smooth vertical profiles of height tendencies, because the tendencies at neighbouring levels are bound to each other by the vertical integration in Eq. (14). On the other hand, our method differs from earlier applications of the Zwack-Okossi equation (e.g. Zwack and Okossi, 1986; Lupo and Smith, 1998) because the use of the generalized omega equation eliminates vertical motion as an indepenent height tendency forcing. This is an important advantage, because these earlier studies have shown a tendency of

compensation between vertical motions and the other forcing terms.

Earlier diagnostic tools have come close to our technique. The most similar approach is probably used in the DIONYSOS (Caron et al., 2006) tool. Regardless of having many similarities, there are still three major differences. First, DIONYSOS eliminates the ageostrophic vorticity tendency as an independent forcing using an iterative procedure. Second, DIONYSOS uses simple parametrizations of diabatic heating and friction, whereas OZO relies directly on model output. Third, DIONYSOS

uses the method of Räisänen (1997) to convert vorticity tendencies to height tendencies, whereas the Zwack-Okossi method is used in OZO.

### 2.3 Vorticity and temperature advection by non-divergent and divergent winds

Following the Helmholtz theorem, the horizontal wind can be divided to non-divergent ($\boldsymbol{V_\psi}$) and divergent ($\boldsymbol{V_\chi}$) parts. Their contributions to vorticity advection and temperature advection can be separated as

$$-\boldsymbol{V}\cdot\nabla\left(\zeta + f\right) = -\boldsymbol{V_\psi}\cdot\nabla\left(\zeta + f\right) - \boldsymbol{V_\chi}\cdot\nabla\left(\zeta + f\right) \tag{22}$$

$$-\boldsymbol{V}\cdot\nabla T = -\boldsymbol{V_\psi}\cdot\nabla T - \boldsymbol{V_\chi}\cdot\nabla T \tag{23}$$

and the same applies to the corresponding $\omega$ and height tendency contributions. This separation between $\boldsymbol{V_\psi}$ and $\boldsymbol{V_\chi}$ contributions is included in OZO because Räisänen (1997) found it to be important for the height tendencies associated with vorticity

advection.

OZO calculates the divergent part of the wind ($\boldsymbol{V_\chi}$) from the gradient of the velocity potential $\chi$ (Eq. 24), which is derived from the horizontal divergence by inverting the Laplacian in Eq. (25).

$$\boldsymbol{V_\chi} = \nabla\chi \tag{24}$$





$$\nabla^2 \chi = \nabla \cdot \boldsymbol{V} \tag{25}$$

The non-divergent wind is obtained as the difference between $\boldsymbol{V}$ and $\boldsymbol{V_\chi}$. OZO output explicitly includes $\omega$ and height tendency contributions of vorticity advection and temperature advection by the full wind field $\boldsymbol{V}$ and the corresponding contributions associated with the divergent wind $\boldsymbol{V_\chi}$. The contributions associated with the non-divergent wind $\boldsymbol{V_\psi}$ can be calculated as their residual.

## 3 Numerical methods

The first version of the OZO software package is tailored to use output from WRF simulations in idealized Cartesian geometry. The software domain is periodic in the zonal direction whereas symmetric boundary conditions are used at the northern and southern boundaries. Before the calculation, the WRF data needs to be interpolated to pressure coordinates.

### 3.1 Calculation of right-hand-side terms

All of the right-hand-side terms of the omega equation (Eq. 5) and the Zwack-Okossi equation (Eq. 16) are calculated in grid point space. Horizontal and vertical derivatives are approximated with two-point central differences with the exception at the meridional and vertical boundaries, where one-sided differences are used. In the calculation of the imbalance term of the omega and Zwack-Okossi equations (Eq. 11 and 21), also tendencies of $T$, $\zeta$ and $Z$ are needed. These tendencies are approximated by two-hour central differences of the corresponding variables.

Because the calculations are done in pressure coordinates, the lower boundary of the domain does not correspond to the actual surface. To mitigate the impact of this, vorticity and temperature advection, friction, diabatic heating and the ageostrophic vorticity tendency are all attenuated below the actual surface by multiplying them by a factor varying from zero to one. The value of the multiplication factor at each level depends on how far down the level is below the ground. For example, for a surface pressure of 950 hPa and vertical mass-centered grid spacing of 50 hPa, the multiplication factor is zero at 1000 hPa, 0.5 at 950 hPa, and one at 900 hPa and all higher levels.

### 3.2 Inversion of left-hand-side operators

The omega equation is solved using a multigrid algorithm (Fulton et al., 1986). Each multigrid cycle starts from the original (finest) grid, denoted below with the superscript (1). A rough solution in this grid ($\tilde{\omega}^{(1)}$) is found using $\nu_1$ iterations of simultaneous underrelaxation, starting either from the previous estimate of $\omega$ or (in the first cycle) $\omega = 0$. The residuals

$$Res^{(1)} = F - L(\tilde{\omega}^{(1)}) \tag{26}$$





are then upscaled to a coarser grid (superscript (2)), in which the number of points is halved in all three dimensions. In this grid, the equation

$$L(\omega^{(2)}) = Res^{(1)} \tag{27}$$

is then roughly solved replicating the method used in the first grid. The residual of this calculation is fed to the next coarser grid, and the procedure is continued until the grid only has five points on its longest (meridional) axis. Thus, for an idealized baroclinic wave simulation with horizontal resolution of 100 km, the meridional axis has 80 grid points. That makes four coarser resolutions (40, 20, 10, and 5 points on the meridional axis) in addition to the original one.

Having obtained the estimate $\tilde{\omega}^{(N)}$ for the coarsest grid, a new estimate for the second coarsest grid is formed as

$$\tilde{\omega}_{NEW1}^{(N-1)} = \tilde{\omega}^{(N-1)} + \alpha \tilde{\omega}^{(N)} \tag{28}$$

where $\alpha$ is a relaxation coefficient. To reduce the effect of regridding errors, this estimate is refined using $\nu_2$ iterations of simultaneous underrelaxation. The result, $\tilde{\omega}_{NEW2}^{(N-2)}$, is then substituted to the next finer grid

$$\tilde{\omega}_{NEW1}^{(N-2)} = \tilde{\omega}^{(N-2)} + \alpha \tilde{\omega}_{NEW2}^{(N-1)} \tag{29}$$

and the sequence is repeated until the original grid is reached.

After each multigrid cycle, the maximum difference between the new and the previous estimate of $\omega$ in the original (finest) grid is computed. If this difference exceeds a user-defined threshold (by default $5 \times 10^{-5}\ \mathrm{Pa\,s^{-1}}$), the multigrid cycle is repeated. Typically, several tens of cycles are required to achieve the desired convergence.

OZO has four parameters for governing the multigrid algorithm, with the following default values: the underrelaxation coefficient ($\alpha = 0.2$), the number of sub-cycle iterations in the descending ($\nu_1 = 4$) and ascending phases of the multigrid cycle ($\nu_2 = 2$), and the threshold for testing convergence ($toler = 5 \times 10^{-5}\ \mathrm{Pa\,s^{-1}}$). All these parameters can be adjusted via a namelist. Note that the mentioned default values of $\alpha$, $\nu_1$ and $\nu_2$ have been optimized for a $100\,\mathrm{km}$ grid resolution. At higher resolution, $\alpha$ may need to be reduced and $\nu_1$ and $\nu_2$ increased to ensure the convergence of the iteration.

In the Zwack-Okossi equation, geostrophic vorticity tendencies are converted to geopotential height tendencies using Eq. (13), which is a two-dimensional Poisson's equation. To solve this equation computationally effectively, we utilize Intel's MKL (Math Kernel Library) Fast Poisson Solver routines, which employ the DFT (Discrete Fourier Transform) method. Intel's MKL is widely spread and freely downloadable (see Section 8), although a registration is required.

### 3.3 Boundary conditions

In the omega equation, homogeneous boundary conditions ($\omega = 0$) at both the meridional and the lower and the upper boundaries are used. For the Zwack-Okossi equation, a slightly more complicated procedure is used to ensure that the area means of the individual height tendency components are consistent with the corresponding temperature tendencies. First, for all of V, T, F, Q and A, the height tendency is initially solved from Eq. (13) using homogenous boundary conditions ($\frac{\partial Z}{\partial t} = 0$) at the northern and southern boundaries. Then, the area mean temperature tendencies for these five terms are calculated, taking





into account both the direct effect of temperature advection (for T) and diabatic heating (for Q) and the adiabatic warming / cooling associated with the corresponding omega component. These temperature tendencies are substituted to the hypsometric equation to calculate the corresponding area mean height tendencies. In the vertical integration of the hypsometric equation, the area mean height tendency at the lower boundary (1000 hPa) is set to zero, following the expectation that the total atmospheric

mass in the model domain is conserved. Horizontally homogeneous adjustments are then made to the initial height tendencies for V, T, F, Q and A, to force their area means to agree with those derived from the hypsometric equation.

## 4  The WRF setup

WRF is a non-hydrostatic model and can generate atmospheric simulations using real data or idealized simulations (Wang et al., 2007; Shamarock et al., 2008). The calculations presented in this paper used input data from an idealized moist baroclinic wave

simulation, which simulates the evolution of a baroclinic wave within a baroclinically unstable jet in the Northern Hemisphere under the f-plane approximation (Blázquez et al., 2013). The value of the Coriolis parameter was set to $10^{-4}\,\mathrm{s}^{-1}$ in the whole model domain.

The simulation was run for ten days with one-hour output interval, in a domain of 4000 x 8000 x 16 km, with 100 km horizontal resolution and periodic boundary conditions in the zonal direction. The horizontal grid was Cartesian and staggered,

and 64 sigma levels were used in the vertical direction. After running the simulation, data was interpolated from model levels to 19 evenly spaced pressure levels (1000, 950, ..., 100 hPa). The interpolation was done with WRF utility *wrf_interp*, which is freely available from the WRF website (see Section 8). During the interpolation, the horizontal data grid was unstaggered to mass points, thus having 40 grid points in the zonal and 80 grid points in the meridional direction.

The model output data contained all the variables needed in solving the generalized omega equation and the Zwack-Okossi

equation: temperature, wind components, geopotential height, surface pressure and parametrized diabatic heating and friction components. Diabatic heating and friction in the WRF included contributions from various physical processes, such as cumulus convection, boundary layer physics and microphysics. The physical tendencies are not in the default WRF output, and need to be added by modifying the WRF Registry file. The Radiation scheme was switched off during the simulation.

## 5  Computational aspects

OZO can be run on a basic laptop with Linux environment, provided that standard NetCDF library, Intel's MKL and some Fortran compiler, preferably GNU's gfortran, are available. The source code of the OZO is written in Fortran 90 standard and can be currently compiled only for a serial version.

The inversion of the left-hand-side operator of the omega equation (Eq. 6) is computationally quite a heavy process. In our previously described test runs, the calculation of the all five $\omega$ components took approximately 2.3 seconds per timestep by

a basic laptop. For the height tendencies, the inversion of the horizontal Laplacian (Eq. 13) is much more straightforward to do, and moreover, the solving is employed by MKL Fast Poisson Solver routines. Hence the corresponding time for height





tendency calculation was only 0.05 seconds. Although these numerical values depend to some extent on the used hardware, they illustrate how the majority of the computing time is currently used by the omega equation solver. The goal is to improve the computational performance by utilizing better scalable solver routines for the omega equation in a future version of the software.

## 6 Results - vertical motion

### 6.1 Comparison between calculated and WRF-simulated vertical motions

Figure 1 compares the solution of the generalized omega equation ($\omega_{TOT}$) with $\omega$ as obtained directly from the WRF output ($\omega_{WRF}$), at the $700 \, \mathrm{hPa}$ level after 118 hours of simulation when the cyclone is approaching its maximum intensity. The agreement is excellent. A strong maximum of rising motion ($\omega \approx -2 \, \mathrm{Pa\,s}^{-1}$) occurs near the occlusion point to the east of the surface low in both cases, with somewhat weaker ascent along the cold frontal zone to the southwest and in the northeastern sector of the low. Descent prevails further east of the low and behind the cold front. However, lots of mesoscale details are visible in both the simulated and the calculated $\omega$, with some although not perfect agreement between these two. The difference between $\omega_{Tot}$ and $\omega_{WRF}$ (Fig. 1c) is noisy, although it suggests a slight positive (downward) bias in $\omega_{TOT}$ to the east of the surface low.

The QG omega equation (Eq. 1) also captures the large-scale patterns of rising and sinking motion reasonably well (Fig. 1d). However, it substantially underestimates the ascent to the east of the low and, in particular, along the cold front. This behavior probably results from the absence of diabatic processes. Furthermore, many of the mesoscale details shared by $\omega_{WRF}$ and $\omega_{TOT}$ are missing in the QG solution.

A more comprehensive statistical evaluation of the calculated vertical motions is given in Figs. 2 and 3, by using data from the whole model area and the eight last days of the simulation. The first two days, when both the cyclone and the vertical motions are still weak, are neglected. Fig. 2 shows the time-averaged spatial correlation between $\omega_{WRF}$ and various omega equation solutions. The correlation between $\omega_{TOT}$ (line $VTFQA$) and $\omega_{WRF}$ is excellent, reaching 0.97 in the midtroposphere and exceeding 0.85 at all levels from 250 to 900 hPa. However, leaving out $\omega_A$, which requires non-synoptic information from the time derivatives of temperature and vorticity, deteriorates the correlation substantially (line $VTFQ$). The solution of the QG omega equation only correlates with $\omega_{WRF}$ at $r \sim 0.7$ in the midtroposphere (line $VT(QG)$), although the correlation approaches 0.8 at 300 hPa where diabatic heating is less important than at lower levels.

Fig. 3 compares the root-mean-square (RMS) amplitudes of $\omega_{WRF}$ and $\omega_{TOT}$. RMS($\omega_{TOT}$) is typically about 10% smaller than RMS($\omega_{WRF}$). This is presumably due to the truncation errors that occur when the derivatives in the forcing terms are approximated with second-order central differences. The RMS amplitudes of the individual $\omega$ components will be discussed in the next subsection.



## 6.2 Contributions of individual forcing terms to $\omega$

Figure 4 shows the contributions of the five individual forcing terms to $\omega(700\,\mathrm{hPa})$ for the situation studied in Fig. 1. Vorticity advection and thermal advection both contribute substantially to the vertical motions (Fig. 4a-b), but the maxima of ascent and descent are both further east for $\omega_T$ (Fig. 4b) than $\omega_V$ (Fig. 4a). Due to this phase shift, there is a cancellation between rising motion from $\omega_V$ and sinking motion from $\omega_T$ just behind the cold front. This cancellation effect is well-known and typically occurs behind the cold front, where cold advection and increasing cyclonic vorticity advection with height take place (Lackmann, 2011). On the other hand, these two terms both induce rising motion to the northeast of the centre of the low. Diabatic heating, which is dominated by latent heat release, strongly enforces the ascent along the main frontal zones of the cyclone (Fig. 4d). Compensating subsidence prevails in the surrounding areas, except for spots of localized ascent associated with convective precipitation well behind the cold front. The imbalance term $\omega_A$ is also not negligible (Fig. 4e), although smaller than $\omega_V$, $\omega_T$ and $\omega_Q$. Friction induces ascent close to the centre of the low and descent around and to the northeast of the surface high (Fig. 4c), but its contribution is quite weak at the $700\,\mathrm{hPa}$ level.

In terms of the RMS amplitudes evaluated over the whole model area and the last 8 days of the simulation, temperature advection makes the largest contribution to the calculated $\omega$ (line $T$ in Fig. 3). However, temperature advection is rivalled by vorticity advection ($V$) in the upper midtroposphere ($\sim 350 - 600\,\mathrm{hPa}$). Diabatic heating ($Q$) and the imbalance term ($A$) are also substantial, particularly in the mid-to-lower troposhere. RMS($\omega_F$) is at its maximum near the top of the boundary layer at $850\,\mathrm{hPa}$ but remains modest even at this level. These results are mostly consistent with similar calculations made for observation-based analysis (Räisänen, 1995) and for model data (Stepanyuk et al., 2016). However, the imbalance term was relatively small in Räisänen's (1995) study, presumably because the 6-hour time resolution of its input data was not sufficient for a proper estimation of this term. To counteract this issue, 1-hour time resolution was used for this paper as well as in Stepanyuk et al. (2016). Conversely, RMS($\omega_F$) is smaller in Fig. 3 than found for the midlatitudes in Räisänen (1995) and Stepanyuk et al. (2016). This may be at least in part because Räisänen (1995) and Stepanyuk et al. (2016) included both land and sea areas, whereas our WRF simulation was made for an idealized ocean surface.

A further division of $\omega_V$ and $\omega_T$ to contributions from advection by rotational and divergent wind reveals that they both are largely dominated by the rotational wind (not shown).

## 7 Results - height tendency

### 7.1 Comparison between calculated and WRF-simulated height tendencies

In this subsection, the calculated total height tendencies are compared with height tendencies from the WRF simulation. The latter ones were estimated as two-hour central differences from the simulated geopotential heights.

Figure 5 shows the distributions of the calculated height tendency, the WRF height tendency and their difference slightly before the cyclone reaches its maximum intensity (t=118 hours). The values are shown at $900\,\mathrm{hPa}$ level, which is sufficiently low to represent the processes affecting the low-level cyclogenesis. Negative (positive) height tendency over the low centre





indicates deepening (weakening) of the low. In general, a close agreement between the fields is seen, but somewhat larger differences occur along the cold front. To study the origin of these differences, we solved the Zwack-Okossi equation (Eq. 14) using the temperature and vorticity tendencies from the WRF simulation. Because the differences in the cold front area remained largely the same as in Fig. 5c, we suspect that they are related to non-hydrostatic effects that are neglected in the
Zwack-Okossi equation.

Figure 6 shows the time-averaged RMS error and correlation coefficient between the calculated and WRF height tendency as a function of height. The RMS error is quite constant up to the 250 hPa level (Fig. 6a). Above this level, the error grows rapidly towards the stratosphere. This error growth is accompanied by a decrease of correlation coefficient at the same altitude. This deterioration is presumably at least partly due to the 50 hPa vertical resolution in the OZO, which is too coarse for an
adequate representation of stratospheric dynamics.

The correlation between the calculated and WRF height tendency is highest in the upper troposphere, being there roughly 0.98. The correlation weakens slightly closer the surface, but still exceeds 0.95. Thus, the calculated height tendency is generally in very good agreement with the tendency diagnosed directly from the WRF output.

## 7.2   Contributions of individual terms at mature stage

The contributions of the individual height tendency components at the 900 hPa level at 118 hours are shown in Fig. 7. Vorticity advection (Fig. 7a) produces a wide and strongly positive height tendency behind the surface low (see Section 7.4 for further analysis of this term). Thermal advection (Fig. 7b) causes a positive height tendency in the area behind the surface low and a negative height tendency at the opposite side. This large negative height tendency ahead of the low is caused by warm air advection in the mid- and upper troposphere. In this baroclinic life cycle simulation, thermal advection is the main contributor
to the movement of the cyclone, which is in agreement with the study of Räisänen (1997).

Friction (Fig. 7c) always acts to damp synoptic-scale weather systems and is thereby inducing a positive (negative) height tendency over the surface low (high). In contrast to friction, diabatic heating (Fig. 7d) is causing uniformly negative lower tropospheric height tendencies in the vicinity of the surface low. As its contribution is also negative at the centre of the surface low, this indicates that diabatic heating plays an essential role in the deepening the cyclone. The largest negative height tendency
due to diabatic heating is located south-east from the low centre, where strong latent heat release occurs in connection with frontal precipitation.

The imbalance term (Fig. 7e) shows more small-scale structure than the other terms. In general, however, it is in phase with the total height tendency near the centre of the low, with negative values to the east and positive values to the west. The reason for this feature is most probably the following: The conversion from geostrophic vorticity tendencies to height tendencies for
the other terms was done by assuming a geostrophic balance according to Eq. (13). However, in cyclones the wind is typically subgeostrophic (e.g. Holton and Hakim, 2012). Therefore, the tendency of geostrophic vorticity exceeds the actual vorticity tendency. This implies that height tendencies calculated from the actual vorticity tendency under the geostrophic assumption will be too small. The imbalance term takes care of this and makes the calculated total height tendency to correspond better to the actual change of the geopotential height field.





### 7.3 Height tendencies at cyclone centre

Figure 8 shows the $900\,\mathrm{hPa}$ level height tendencies induced by the five individual terms in the cyclone centre during the deepening period. The low deepens vigorously roughly between 72 and 120 hours of simulation, as shown by the negative total height tendency (black line) during this period. The total height tendency is also in a good agreement with the WRF height tendency (dotted line), although some positive bias is seen in the end of the period.

The deepening is mostly due to vorticity advection (blue line) and the imbalance term (grey line). Later on, roughly from 120 hours onward, as the cyclone reaches its mature stage, diabatic heating (magenta line) and thermal advection (orange line) make the largest contribution to maintaining the intensity of the surface low. Friction (green line) systematically destroys cyclonic vorticity over the cyclone centre and thus produces positive height tendency during the whole life cycle. The effect of friction becomes larger as the intensity of the cyclone increases.

### 7.4 The effect of vorticity and thermal advection by divergent and nondivergent winds

Figure 9 presents the height tendencies associated with vorticity advection by $\boldsymbol{V}_\chi$ and $\boldsymbol{V}_\psi$ separately. The divergent wind vorticity advection causes widespread and strong positive height tendency over and around the surface low (Fig. 9a). According to Räisänen (1997), the divergent wind transports anticyclonic vorticity from the surroundings of the surface low, and is thus acting to reduce the cyclonic vorticity at the centre of the low. In the case of the nondivergent wind component (Fig. 9b), positive (negative) height tendencies behind (ahead) of the low originate from the upper troposphere, where the nondivergent wind and thereby vorticity advection is the strongest. Cyclonic vorticity advection ahead of the trough produces negative height tendency in the same area, while anticyclonic vorticity advection ahead of the ridge does the opposite. Furthermore, the nondivergent vorticity advection is substantially contributing to the deepening of the cyclone, since the area of the negative height tendency reaches the centre of the low as well.

In contrast to vorticity advection, thermal advection by divergent winds was found to cause a negligible height tendency (not shown).

## 8 Conclusions

In this paper, a software package called OZO is introduced. OZO is a tool for investigating the physical and dynamical factors that affect atmospheric vertical motions and geopotential height tendencies, tailored for WRF simulations with idealized Cartesian geometry. As input to OZO, the standard output of the WRF model interpolated to evenly spaced pressure levels is required.

The generalized omega equation diagnoses the contributions of different physical and dynamical processes to vertical motions: vorticity advection, thermal advection, friction, diabatic heating, and imbalance between temperature and vorticity tendencies. Then, analogously with the vertical motion, the height tendencies associated with these forcings are calculated. As an advance over traditional applications of the Zwack-Okossi equation (Zwack and Okossi, 1986; Lupo and Smith, 1998), the





use of the generalized omega equation allows OZO to eliminate vertical motion as an independent forcing in the calculation of height tendencies.

The calculated total vertical motions and height tendencies in the test case are generally in excellent agreement with the vertical motions and height tendencies diagnosed directly from the WRF simulations. The time-averaged correlation between

the calculated and the WRF height tendency was 0.95-0.98 in the troposphere. For the vertical motion as well, a correlation of 0.97 was found in the midtroposphere. Our analysis further illustrates the importance of both adiabatic and diabatic processes to atmospheric vertical motions and the development of the simulated cyclone.

The OZO software is applicable to different types of WRF simulations, as far as Cartesian geometry is used. One example of potential applications are simulations with increased sea surface temperatures as the lower boundary condition. Combined

with OZO, such simulations provide a simple framework for studying the changes in cyclone dynamics in a warmer climate.

## 9   Data and code availability

The source code of the OZO is freely available under MIT license in GitHub (https://github.com/mikarant/ozo). OZO v.1.0 described in this manuscript is also archived at http://doi.org/10.5281/zenodo.157188. In addition to the source code, the package includes also a makefile for compiling and running the program, a small sample input dataset for testing the function-

ality and two README-files containing the instructions for both generating input data with WRF and running the OZO program. The WRF model as well as the interpolation utility (*wrf_interp*) are downloadable from the WRF users page (http://www2.mmm.ucar.edu/wrf/users/downloads.html). Intel's MKL can be downloaded after registration from their web page (https://software.intel.com/en-us/articles/free-mkl). OZO v1.0 is guaranteed to work with WRF version 3.8.1.

## 10   Acknowledgments

We thank the Doctoral Programme of Atmospheric Sciences, University of Helsinki for financially supporting the work of MR. The work of OS was supported by Maj and Thor Nessling foundation (project 201600119) and the work of OR was supported by Vilho, Yrjö and Kalle Väisälä Foundation. VAS was supported by the Academy of Finland Centre of Excellence Program (grant 272041).



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



**Table 1.** List of mathematical symbols.

| | |
|---|---|
| $c_p = 1004$ J kg$^{-1}$ | specific heat of dry air at constant volume |
| $f$ | Coriolis parameter |
| $F$ | forcing in the omega equation |
| $\boldsymbol{F}$ | friction force per unit mass |
| $g = 9.81$ m s$^{-2}$ | gravitational acceleration |
| $\boldsymbol{k}$ | unit vector along the vertical axis |
| $L$ | linear operator in the left-hand-side of the omega equation |
| $p$ | pressure |
| $Q$ | diabatic heating rate per mass |
| $R = 287$ J kg$^{-1}$ | gas constant of dry air |
| $S = -T\frac{\partial ln\theta}{\partial p}$ | stability parameter in pressure coordinates |
| $t$ | time |
| $T$ | temperature |
| $\boldsymbol{V}$ | horizontal wind vector |
| $\boldsymbol{V_g}$ | geostrophic wind vector |
| $\alpha$ | relaxation coefficient |
| $\sigma = -\frac{RT}{p\theta}\frac{\partial \theta}{\partial p}$ | hydrostatic stability |
| $\sigma_0$ | isobaric mean of hydrostatic stability |
| $\zeta$ | vertical component of relative vorticity |
| $\zeta_g$ | relative vorticity of geostrophic wind |
| $\zeta_{ag}$ | relative vorticity of ageostrophic wind |
| $\omega = \frac{dp}{dt}$ | isobaric vertical motion |
| $\nabla$ | horizontal nabla operator |
| $\nabla^2$ | horizontal laplacian operator |





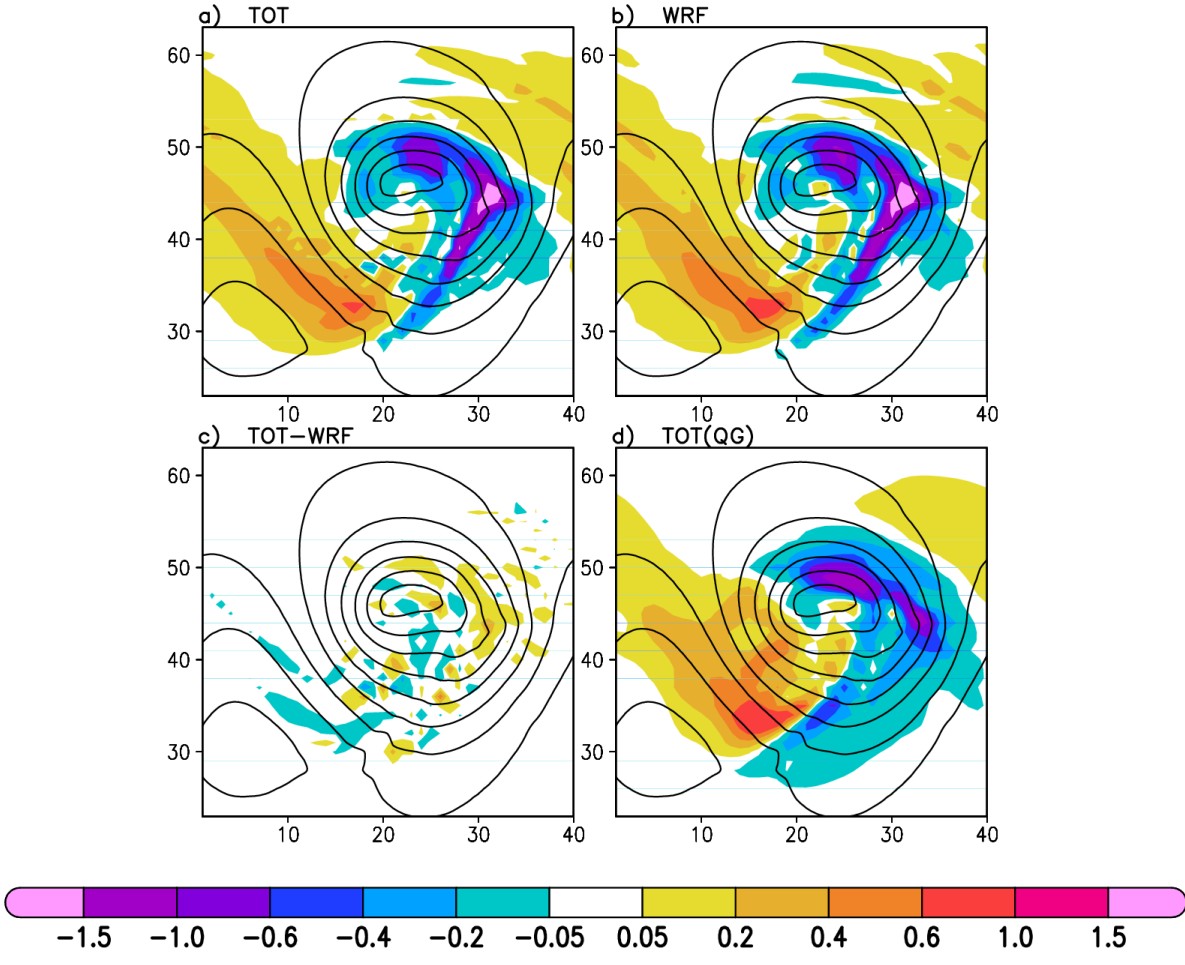

**Figure 1.** a) The sum of all $\omega$ components from Eq. (5) ($\omega_{TOT}$), b) $\omega$ from WRF ($\omega_{WRF}$), c) difference ($\omega_{TOT} - \omega_{WRF}$) and d) solution of the QG omega equation ($\omega_{V(QG)} + \omega_{T(QG)}$) at 700 hPa level at time 118 h. Unit is $\mathrm{Pa\,s^{-1}}$. Contours show geopotential height at 900 hPa level with an interval of 50 m. Labels on x- and y-axes indicate grid point numbers. Note that the area covers only half of the model domain in the meridional direction.





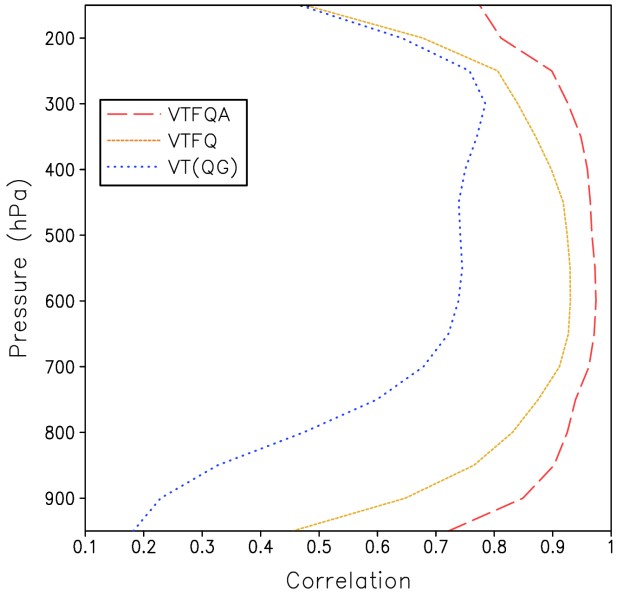

**Figure 2.** Correlation of the omega equation solutions with $\omega_{WRF}$. $VTFQA = \omega_{TOT}$, $VTFQ = \omega_{TOT} - \omega_A$, $VT(QG) = \omega_{V(QG)} + \omega_{T(QG)}$.

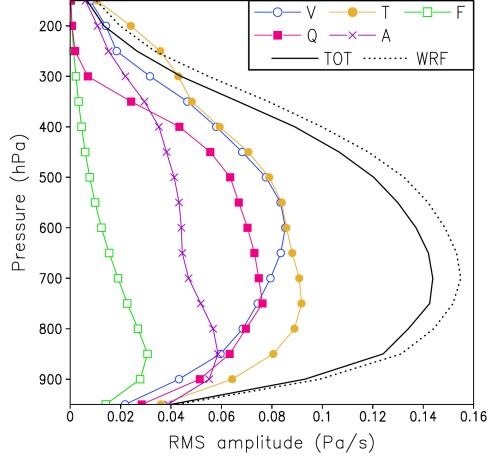

**Figure 3.** RMS amplitudes of $\omega_{WRF}$, $\omega_{TOT}$, and the individual $\omega$ components from (5).


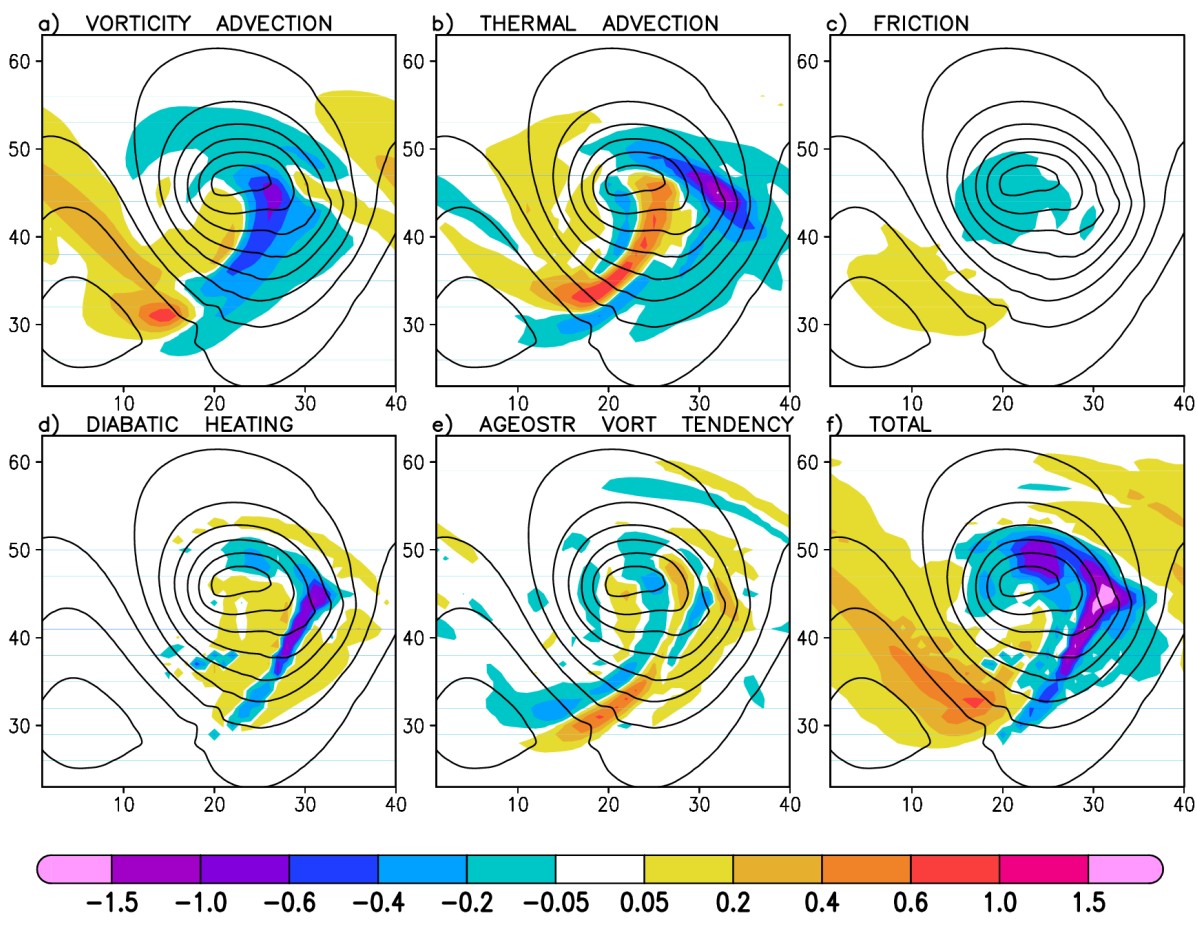

**Figure 4.** Vertical motions induced by individual forcing terms at level 700 hPa at time 118 h. a) $\omega_V$, b) $\omega_T$, c) $\omega_F$, d) $\omega_Q$, e) $\omega_A$ and f) $\omega_{TOT}$. Unit is $\mathrm{Pa\,s}^{-1}$. and contour lines show 900 hPa geopotential height with 50 m interval.



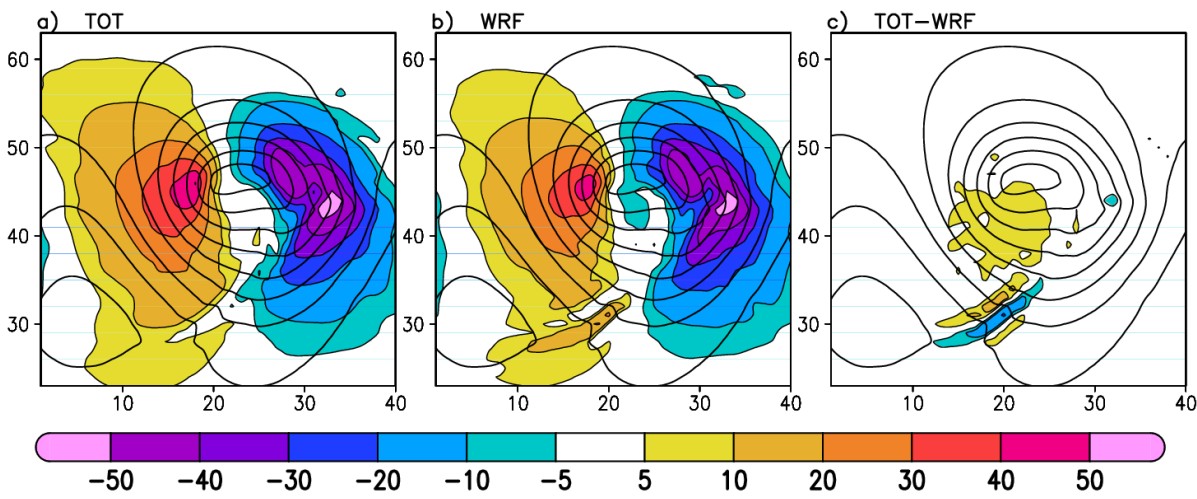

**Figure 5.** As Fig. 1, but for height tendency at 900 hPa. Unit is m $(2\mathrm{h})^{-1}$.

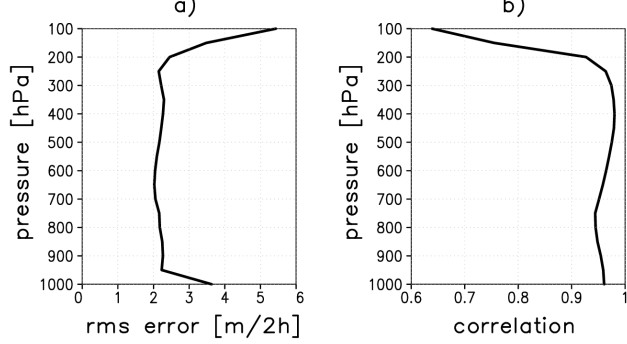

**Figure 6.** Time mean a) rms error and b) spatial correlation coefficient between calculated and WRF height tendency over the last 8 days of the 10-day simulation.





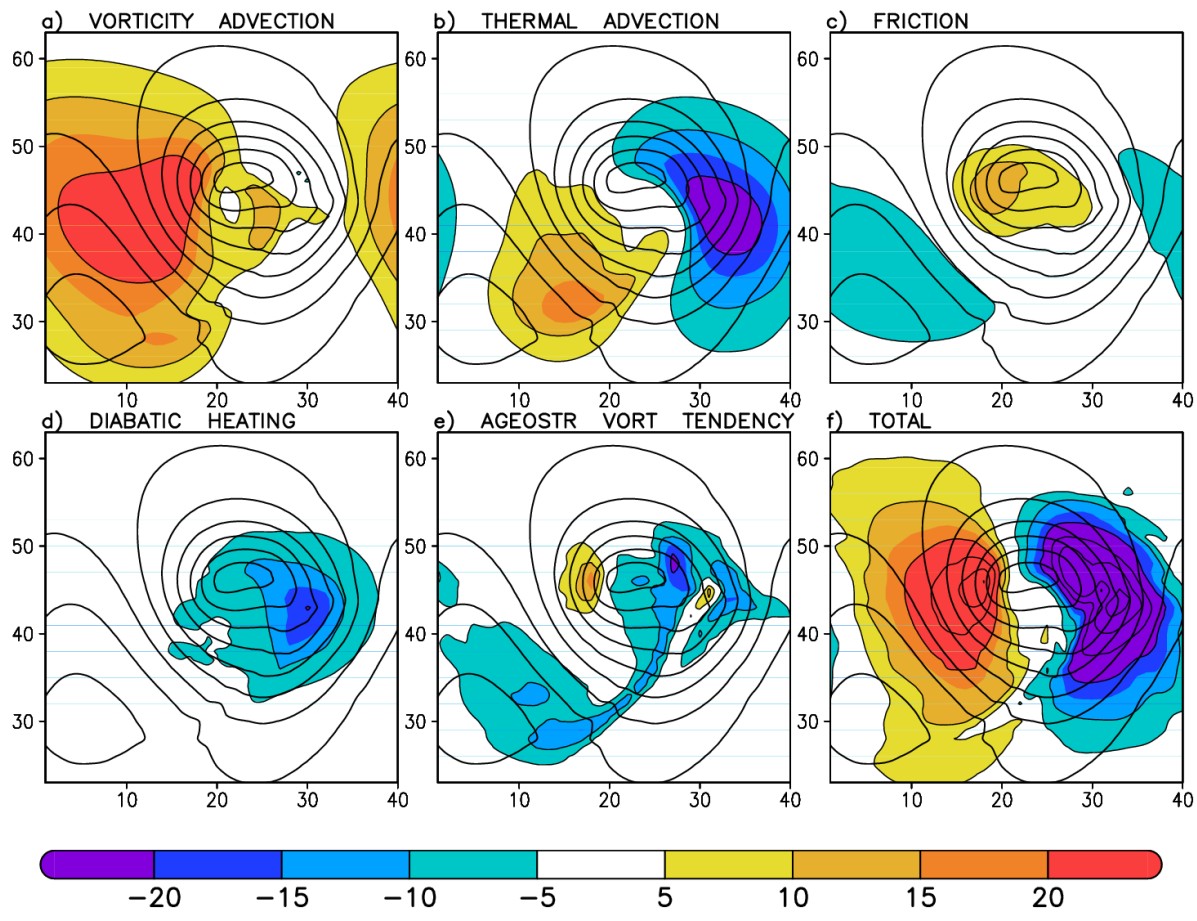

**Figure 7.** As Fig. 4, but for height tendency components at 900 hPa level. Unit is $\mathrm{m\,(2h)}^{-1}$. Note that the color scale of f) differs from Fig. 5a.





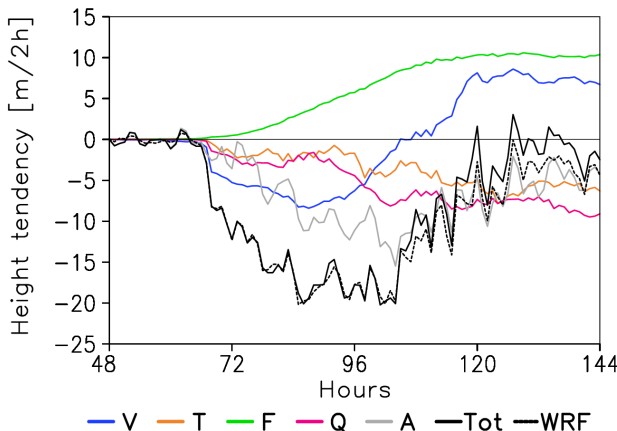

**Figure 8.** Time series of individual height tendency components at the 900 hPa level from the cyclone centre during the deepening period. V = vorticity advection, T = thermal advection, F = friction, Q = diabatic heating, A = imbalance term, Tot = total and WRF = WRF height tendency. Height tendencies at the cyclone centre were averaged over all grid boxes in which the 900 hPa geopotential height was less than 10 m above its minimum.

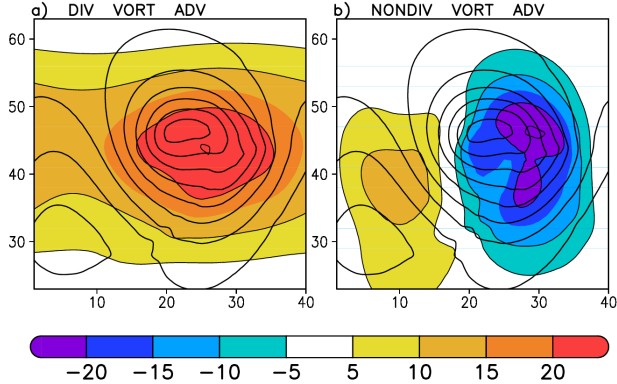

**Figure 9.** Height tendency associated with vorticity advection by the a) divergent and b) nondivergent winds at 900 hPa at time 118 h. Unit is $\mathrm{m\,(2h)^{-1}}$.