# Peer review of "OZO v.1.0: Software for solving a generalized omega equation and the Zwack-Okossi height tendency equation using WRF model output"

_Geoscientific Model Development, 2016_

## Referee Comment (RC1) · Anonymous Referee #1 · 26 Oct 2016

GENERAL COMMENTS:

Although the software described here does not provide new or recently developed diagnostic methods to potential end-users, it proposed a complete set of diagnostic equations that could be very useful to the WRF community.

My only concern is that, given the information provided in the paper, the software seems adapted only to numerical weather prediction at low resolution ($\sim 10^2$km), which is likely to lower considerably the interest in a modelling community focusing nowadays on kilometric-scale ($\sim 10^0$km) and sub-kilometric-scale ($\sim 10^{-1}$km) applications. I do not consider this to be a showstopper for the publication of this paper but the authors needs to explain more clearly the application limits of their software and

why, given these limitations, it is still a relevant tool.

Otherwise, this is a well-written and well-organized paper.

MAJOR COMMENTS:

1. There are two main reasons why this software does not seems suitable for today's state-of-the-art limited-area applications: the hydrostatic assumption and the choice of the computational grids. The hydrostatic assumption, made in the derivation of the diagnostic equations, is probably not as dramatic given that non-hydrostatic phenomena are, in many cases, a secondary factor in the atmospheric flow. However, adopting horizontal and vertical computational grids different than WRF's computational grids is likely to introduce a significant amount of errors and noises, particularly in the smallest scales. It is the responsibility of the authors to state the limits of application of their software or to provide evidence that the above potential problems can actually be overcome by their software.

2. Even in the idealized simulation presented in the paper using a 100km horizontal grid configuration, numerical errors and noises should be generated by the software. The figures presented in this paper show fields that are generally smooth, but all the chosen fields are resulting from the inversion of a Laplacian operator, which project mostly on the largest scales, thus hiding the small-scale numerical error and noises. Do the authors use a filtering strategy in the software to alleviate this issue? It would be worthwhile to show some forcing fields (like vorticity and temperature advection). In order to study the dynamics of the atmosphere, looking at the forcing themselves is as important as looking at their contribution to different variables.

MINOR COMMENTS:

1. P.3 L.8-9: It is important to stress the role of the Laplacian operator in (4). Warm (cold) advections do not directly imply rising (sinking) motion; it is the horizontal distribution of the advection that matters. Therefore, I suggests the following modification to

the text: '. . . with height and a maximum of warm (cold) advection should induce rising (sinking) motion'

2. P.8, Section 3.3: Have you considered taking into the account the orographic forcing? Otherwise, this is likely to reduce considerably the performances of the software over complex terrain. . .

3. P. 12. L. 4-5: It is unlikely that non-hydrostatic effects where important in this simulation given the low horizontal resolution employed (at this resolution the deep convection scheme should be active well before a column of the simulated atmosphere becomes truly unstable). Given the relative small-scale of the differences, numerical errors are likely to be the main source of discrepancy.

―――――――――――――――――――

---

## Referee Comment (RC2) · Anonymous Referee #2 · 28 Oct 2016

General comments The authors present OZO, a diagnostic tool for numerical atmospheric models that calculates the vertical motions associated with different physical forcings and the corresponding height tendencies with the help of the generalized omega and the Zwack-Okossi tendency equations. With OZO it is possible to calculate the vertical velocity and height forcing for each physical mechanism, including both the direct effect from the forcing itself and the indirect effects related to the vertical motion induced by each physical mechanism. Another interesting feature of the software is the possibility to infer the vertical velocity from pressure level data. For the diagnostics OZO uses the hydrostatic primitive omega, vorticity, and thermodynamic equations as well as the nonlinear balance equation. These equations can be solved using standard

output and makes OZO potentially useful for the diagnostic of any atmospheric model. The paper is well written, with a concise and complete abstract and the overall presentation well structured and clear. The description of the equations is complete, with correctly defined mathematical formulae and the methods and the assumptions made are valid and clearly outlined. The references seem appropriate. They give a correct overview of the field and their numbers are adequate. The authors state that the software could be useful for studying the dynamics of such important climate features as storms, low level jets, etc. As such, OZO could be very useful for understanding changes in the dynamics of these phenomena due to changes in the environment, say climate change or the transition of tropical cyclones to extratropical cyclones. However, that kind of studies could be more challenging from the numerical point of view than the case presents in the paper and could need a WRF setup with much higher resolution than the presented here. Therefore I would like to see the performance of OZO with a higher resolution, realistic application (i.e. in a 3-10 km grid and simulations for different seasons). If OZO is able to show good results in such applications, it would represent an important tool for the climate and weather prediction modeling community. Another shortcoming of the software is that it is tailored to use input from WRF, while other similar packages (for instance, DIONYSOS, RIP4) can run on other numerical models. Although I guess it would be not too difficult to extend OZO for the use with the output from other regional models. Although the authors describe similar packages and indicate clearly the differences between OZO and these tools, I miss a more direct comparison of their results.

Specific comments 1. In Räisänen (1995) the method is applied to the ERA reanalysis. Why this time you choose the idealized case and not the real case of Blazquez et al (2012)? Last generation reanalysis are considered significantly better than these available when Räisänen (1995) was published. Would the use of a state of the art reanalysis improve the correlations for the different regions obtained in that paper? 2. The use of the Coriolis parameter f would not cause numerical problems if you analyze a real case? 3. Is your software suited for the analysis of long term climate

simulations? 4. I wonder if the performance of OZO depends on resolution. 5. Model setup description is not clear, should be as in Blazquez et al (2012) 6. I am not sure that at 100 km resolution non-hydrostatic effects are relevant

Technical corrections Introduction, paragraph 5: change "to separate individual forcings contributions to vertical motion and height tendency" to "to separate the contributions of each forcing to the vertical motion and height tendency". Line 8, page 2: "but the division of the $\omega$ to contributions of various atmospheric processes is not possible in RIP4" to "but the division of the $\omega$ tendency into contributions from various atmospheric processes is not possible in RIP4" I do not understand the phrase "OZO output explicitly includes $\omega$ and height tendency contributions of vorticity advection and temperature advection by the full wind field V and the corresponding contributions associated with the divergent wind V$\chi$" In line 3, page 7. May be it could be changed to "OZO output explicitly includes the vorticity advection and temperature advection terms of the $\omega$ and height tendency equations due to both the full wind field V and the the divergent wind V$\chi$"

Please also note the supplement to this comment:
http://www.geosci-model-dev-discuss.net/gmd-2016-219/gmd-2016-219-RC2-supplement.pdf

———————————————

---

## Author Comment (AC1) · 12 Jan 2017

**To:** Anonymous Referee #1

**From:** Mika Rantanen et al.

**Subject:** Author's response for the Referee comments

**Date:** January 12, 2017
* * *
Dear Referees,

We thank you for your constructive reviews on our manuscript. The comments helped us to express more clearly both the applications and the limitations of our software.
In particular, the referee comments identified (i) the coarse resolution and (ii) the idealized geometry assumed by the current version of OZO as major limitations. We agree that applications to high-resolution real-world simulations would be of obvious interest. However,

i) Although generalization of OZO to real-world spherical geometry and realistic geography is possible and we are indeed planning this for future releases, this is a major effort that goes beyond what is possible with our resources in the short run.

ii) Idealized simulations in cartesian geometry are a powerful and commonly used tool in studies of (e.g.) cyclone dynamics, because their simplicity makes it easier to control and understand the influence of different factors. For the same reason, they are useful for education. This, together with the relative simplicity of the numerical implementation, motivated us to tailor the current version of OZO for this idealized geometry.

In the revised manuscript, we discuss these issues more explicitly and have also added to the end of the paper a section on future development possibilities. In addition, we have run WRF, and show results for OZO, at 25 km horizontal resolution which represents a four-fold upgrade from the original manuscript.

In addition, a large number of smaller changes to the manuscript have been made based on the review comments. Since we have made relatively large revisions to the manuscript, we kindly recommend the referees to read it again.

The point-by-point reply to the comments of reviewer #1 is below. Your comments are marked in *italic* and our responses in blue. Furthermore, a marked-up manuscript with tracked changes is attached to the end of this paper.

*General comments*
*Although the software described here does not provide new or recently developed diagnostic methods to potential end-users, it proposed a complete set of diagnostic equations that could be very useful to the WRF community. My only concern is that, given the information provided in the paper, the software seems adapted only to numerical weather prediction at low resolution ($\sim 10^2$ km), which is likely to lower considerably the interest in a modelling community focusing nowadays on kilometric-scale ($\sim 10^0$ km) and sub-kilometric-scale ($\sim 10^{-1}$ km) applications. I do not consider this to be a showstopper for the publication of this paper but the authors needs to explain more clearly the application limits of their software and why, given these limitations, it is still a relevant tool. Otherwise, this is a well-written and well-organized paper.*

We agree that 100 km horizontal resolution is coarse by today's standards, even though we believe that even this would be useful at least for educational purposes. Therefore, the new version of the manuscript represents results for a 25 km resolution simulation which is state-of-the-art in the type of idealized baroclinic wave simulations that is studied in the paper. In principle, OZO can be run at even higher resolution, although the computing time (which mainly goes to solving the generalized omega equation) increases steeply with increasing number of grid points. However, as the equations of OZO were derived from quasi-geostrophic theory (although with many of the QG approximations except for hydrostatic balance relaxed), the interpretation of the results might become difficult at kilometer- or sub-kilometer-scale resolution.

The results at 25 km resolution are discussed in Sections 6 and 7, whereas results at 100 km resolution are presented briefly in the supplementary material of the revised manuscript. Limitations of the software are discussed in Section 8.

**Major comments**

**1.**
*There are two main reasons why this software does not seems suitable for todays state-of-the-art limited-area applications: the hydrostatic assumption and the choice of the computational grids. The hydrostatic assumption, made in the derivation of the diagnostic equations, is probably not as dramatic given that non-hydrostatic phenomena are, in many cases, a secondary factor in the atmospheric flow. However, adopting horizontal and vertical computational grids different than WRFs computational grids is likely to introduce a significant amount of errors and noises, particularly in the smallest scales. It is the responsibility of the authors to state the limits of application of their software or to provide evidence that the above potential problems can actually be overcome by their software.*

The current version of OZO was tailored for the cartesian grid for two reasons: 1) our own scientific interests related to WRF's idealized baroclinic wave simulations and 2) the relative simplicity of the numerical solution of the equations in the cartesian grid. It is our intention to generalize OZO for more realistic grid geometry in further releases of the software, but this is a time-consuming effort which we cannot realistically achieve within the next few months.

Hydrostatic balance is implicitly assumed when isobaric coordinates are used. We also believe that non-hydrostatic effects play a relatively small role in most cases.

These limitations are discussed in Section 8 of the revised manuscript.

**2.**
*Even in the idealized simulation presented in the paper using a 100km horizontal grid configuration, numerical errors and noises should be generated by the software. The figures presented in this paper show fields that are generally smooth, but all the chosen fields are resulting from the inversion of a Laplacian operator, which project mostly on the largest scales, thus hiding the small-scale numerical error and noises. Do the authors use a filtering strategy in the software to alleviate this issue? It would be worthwhile to show some forcing fields (like vorticity and temperature advection). In order to study the dynamics of the atmosphere, looking at the forcing themselves is as important as looking at their contribution to different variables.*

In fact, no filtering of any kind was/is used in OZO or in plotting the maps shown in the paper.

We agree that looking at forcings themselves is important when studying the dynamics of the

atmosphere. For this reason, we added a function to OZO software which writes out forcing fields to the output file if it is wanted. In addition, figures of the forcing fields have been added to the supplementary material (Fig. S1 and S2).

**Minor comments**

**1.**
*P.3 L.8-9: It is important to stress the role of the Laplacian operator in (4). Warm (cold) advections do not directly imply rising (sinking) motion; it is the horizontal distribution of the advection that matters. Therefore, I suggests the following modification to the text: ... with height and a maximum of warm (cold) advection should induce rising (sinking) motion*

We agree with you on this. The text has been modified according to your suggestion in page 3 at lines 10-11.

**2.**
*P.8, Section 3.3: Have you considered taking into the account the orographic forcing? Otherwise, this is likely to reduce considerably the performances of the software over complex terrain...*

The current version of OZO does not include an orograhic forcing term. However, as we note in page 4, at lines 13-14, the effects of orograhic forcing could be mimicked by including an additional term that results from the lower boundary vertical motion as diagnosed directly from WRF.

**3.**
*P. 12. L. 4-5: It is unlikely that non-hydrostatic effects where important in this simulation given the low horizontal resolution employed (at this resolution the deep convection scheme should be active well before a column of the simulated atmosphere becomes truly unstable). Given the relative small-scale of the differences, numerical errors are likely to be the main source of discrepancy.*

This issue is no longer present in our 25 km simulation. The related text was deleted.

[revised manuscript text omitted]

---

## Author Comment (AC2) · 12 Jan 2017

**To:** Anonymous Referee #2

**From:** Mika Rantanen et al.

**Subject:** Author's response for the Referee comments

**Date:** January 12, 2017
* * *
Dear Referees,

We thank you for your constructive reviews on our manuscript. The comments helped us to express more clearly both the applications and the limitations of our software.
In particular, the referee comments identified (i) the coarse resolution and (ii) the idealized geometry assumed by the current version of OZO as major limitations. We agree that applications to high-resolution real-world simulations would be of obvious interest. However,

i) Although generalization of OZO to real-world spherical geometry and realistic geography is possible and we are indeed planning this for future releases, this is a major effort that goes beyond what is possible with our resources in the short run.

ii) Idealized simulations in cartesian geometry are a powerful and commonly used tool in studies of (e.g.) cyclone dynamics, because their simplicity makes it easier to control and understand the influence of different factors. For the same reason, they are useful for education. This, together with the relative simplicity of the numerical implementation, motivated us to tailor the current version of OZO for this idealized geometry.

In the revised manuscript, we discuss these issues more explicitly and have also added to the end of the paper a section on future development possibilities. In addition, we have run WRF, and show results for OZO, at 25 km horizontal resolution which represents a four-fold upgrade from the original manuscript.

In addition, a large number of smaller changes to the manuscript have been made based on the review comments. Since we have made relatively large revisions to the manuscript, we kindly recommend the referees to read it again.

The point-by-point reply to the comments of reviewer #2 is below. Your comments are marked in *italic* and our responses in blue. Furthermore, a marked-up manuscript with tracked changes is attached to the end of this paper.

**General comments**

*The authors state that the software could be useful for studying the dynamics of such important climate features as storms, low level jets, etc. As such, OZO could be very useful for understanding changes in the dynamics of these phenomena due to changes in the environment, say climate change or the transition of tropical cyclones to extratropical cyclones. However, that kind of studies could be more challenging from the numerical point of view than the case presents in the paper and could need a WRF setup with much higher resolution than the presented here. Therefore I would like to see the performance of OZO with a higher resolution, realistic application (i.e. in a 3-10 km grid and simulations for different seasons). If OZO is able to show good results in such applications, it would represent an important tool for the climate and weather prediction modeling community.*

It is clear that applications to real-world cases would be both very interesting and more challenging than the idealized case studied in the paper. It is in our longer-term plan to generalize OZO for use in realistic cases with realistic geometry, but this will require time-consuming work that goes beyond of what we can achieve in the near future. Therefore, the limitations of the current version of the software and avenues for future development are discussed more explicitly in Section 8.

For the revised paper, we have run WRF, and show results for OZO, at 25 km horizontal resolution, which is state-of-the-art in the type of idealized baroclinic wave simulations that is studied in the paper. In principle, OZO can be run at even higher resolution, although the computing time (which mainly goes to solving the generalized omega equation) increases steeply with increasing number of grid points. However, as the equations of OZO were derived from quasi-geostrophic theory (although with many of the QG approximations except for hydrostatic balance relaxed), the interpretation of the results might become difficult at even higher resolution.

The results at 25 km resolution are discussed in Sections 6 and 7, whereas results at 100 km resolution are presented briefly in the supplementary material.

*Another shortcoming of the software is that it is tailored to use input from WRF, while other similar packages (for instance, DIONYSOS, RIP4) can run on other numerical models. Although I guess it would be not too difficult to extend OZO for the use with the output from other regional models.*

We totally agree that the software would gain more users if it was generalized for the most common atmospheric models in addition to WRF. However, with our resources this job would have been too extensive to carry out. Our future goal is to extend OZO to global models, such as OpenIFS, but in this paper we wanted to document the first version of OZO, which is directly applicable only with WRF.

*Although the authors describe similar packages and indicate clearly the differences between OZO and these tools, I miss a more direct comparison of their results.*

We added references to DIONYSOS paper, in which similar kind of RMS - and correlation calculations as in our paper have been done. These comparisons are written in page 10, at lines 11-15, and in page 13, at lines 8-9.

**Specific comments**

**1.**
*In Räisänen (1995) the method is applied to the ERA reanalysis. Why this time you choose the idealized case and not the real case of Blazquez et al (2012)? Last generation reanalysis are considered significantly better than these available when Räisänen (1995) was published. Would the use of a state of the art reanalysis improve the correlations for the different regions obtained in that paper?*

There are several reasons why we chose the idealized WRF simulation and not any real case simulation or reanalysis data:

1) Our own interest to idealized baroclinic wave simulations, which provide a testbed for studying the dynamics of midlatitude weather systems in a simple context.

2) The cartesian geometry in the WRF's idealized barocline wave simulations simplifies the numerical approximations needed in OZO. WRF simulations for real-world cases employ spherical coordinates, which are incompatible with OZO v.1.0. It is our longer-term goal to generalize OZO for realistic geometry but the required work effort goes beyond our short-term possibilities (see the discussion in Section 8).

3) To calculate diabatic heating and friction components in OZO (in both the Omega and Zwack-Okossi equations), parameterized temperature and wind tendency fields are needed. These are not generally available in reanalyses.

**2.**
*The use of the Coriolis parameter f would not cause numerical problems if you analyze a real case?*

We are not sure if we understood the question correctly. However, we would not expect the variation of the Coriolis parameter to cause problems, at least if the domain does not cross the equator. Note that, to ensure consistency, OZO reads the values of the Coriolis parameter directly from WRF output.

**3.**
*Is your software suited for the analysis of long term climate simulations?*

If it would be possible to carry out climate simulations in the geometry used in OZO - then why not. However, the current version of OZO is tailored for Cartesian geometry that cannot be used in global climate models.

**4.**
*I wonder if the performance of OZO depends on resolution.*

To be precise, the resolution is not the limiting factor in OZO performance but the number of grid points. So, if we decrease the grid spacing, we should decrease also the model domain in order to keep the computational performance same. We added Table 2 to the manuscript which shows the dependence of the computing time on the number of grid points. In addition, this issue is discussed more in Sections 5 and 8.

**5.**
*Model setup description is not clear, should be as in Blazquez et al (2012)*

We tried to modify the WRF setup section in the paper more clear. See Section 4.

**6.**
*I am not sure that at 100 km resolution non-hydrostatic effects are relevant*

This problem is no longer present in our simulations. The related text has been deleted.

**Technical corrections**

*Introduction, paragraph 5: change "to separate individual forcings contributions to vertical motion and height tendency" to "to separate the contributions of each forcing to the vertical motion and height tendency".*

Thank you for the comment. The text has been modified according to your suggestion. (Page 2, Lines 8-9)

*Line 8, page 2: "but the division of the $\omega$ to contributions of various atmospheric processes is not possible in RIP4" to "but the division of the $\omega$ tendency into contributions from various atmospheric processes is not possible in RIP4"*

We slightly disagree on this. Precisely said, we don't want to divide $\omega$ tendency ($d\omega/dt$) to contributions - but $\omega$ itself. For that reason the text is changed based on your suggestion, but without the word "tendency". (Page 2, Lines 12-13)

*I do not understand the phrase "OZO output explicitly includes $\omega$ and height tendency contributions of vorticity advection and temperature advection by the full wind field V and the corresponding contributions associated with the divergent wind $V_\chi$" In line 3, page 7. May be it could be changed to "OZO output explicitly includes the vorticity advection and temperature advection terms of the $\omega$ and height tendency equations due to both the full wind field V and the the divergent wind $V_\chi$"*

The sentence has been rephrased - hopefully it is now more understandable (Page 7, Lines 9-11).

[revised manuscript text omitted]

---

## Author Response (AR2)

**To:** Anonymous Referee #2

**From:** Mika Rantanen et al.

**Subject:** Author's response for the Referee's minor comments

**Date:** February 3, 2017
* * *
Dear Referee #2,

We are glad that our major revision to our manuscript has largely satisfied you. We also thank you for your minor comments and hope that our responses will clarify the rest of unclear matters.

The point-by-point reply to your comments is below. Your comments are marked in *italic* and our responses in blue. Furthermore, a marked-up manuscript with tracked changes is attached to the end of this paper.

**General comments**

*The paper has significantly improved. The use of a WRF 25 km setup to showcase OZO gives a better understanding of the strengths of the software and allows to find spots where there is room for improvement. Although most of my comments were addressed, I still miss the analysis of a simulation with a domain with higher horizontal resolution and realistic orography.*

As we noted already in our previous response, the realistic case with realistic orography requires the implementation of spherical coordinates and different handling of the boundary conditions. This would be a major undertaking for us, which we cannot achieve in the near future. However, we have already a draft version of OZO which employs faster and parallelized omega equation solver algorithm. We aim to extend it to global and realistic model simulations in the future but that will be totally outside the scope and timetable of this paper.

**Specific comments**

*1. I do not see how the interpretation of the results might become difficult at higher resolutions.*

At higher resolutions the small-scale temperature tendencies caused by thermal advection and diabatic heating will become more pronounced. However, due to their small temporal and spatial scale, the atmosphere usually does not have time to adjust to the new situation and thus compensating vertical motion does not take place. In our calculations, those situations are taken into account by the imbalance term, which is often canceling out the effect of small-scale temperature tendencies. This is discussed further in the manuscript Section 6.2, and can be also seen in Fig. 5, where the imbalance term (Fig. 5e) has a strong tendency to compensate the vertical motion caused by thermal advection (Fig. 5b).

If we increase the resolution further from the 25 km, we expect to see the increase particularly in the magnitude of thermal advection, diabatic heating and imbalance terms (as you can already see when comparing Fig. 4 and S4). The interpretation of the results will become more difficult as the imbalance term will strongly compensate the effects of these other two terms.

This issue is discussed briefly in Section 8 in the revised manuscript.

*2. Are the symmetric boundary conditions at the meridional boundaries mandatory? Have you tried other boundary conditions? This would be the case for a tipical application.*

The symmetric boundary conditions at the meridional boundaries are not mandatory but strongly recommended. In fact, it is possible to choose other boundaries in WRF, such as periodic or open boundaries, but as there is not really anything happening near the meridional boundaries in our idealized set-up, the change of boundary conditions is not meaningful. For this reason, we have not tried any other boundary conditions.

*3. Line 26, page 8. Given your experience with the 25 and 100 km grids, I would leave for future users the case for both higher and lower resolution.*

We are not totally sure about the meaning of this comment. If you mean that the discussion about results at 100 km horizontal resolution should have also been included to the main manuscript, we didn't want to put it there as
1) it would have made the paper too long and
2) The results at 100 km resolution, and some discussion of them, are available in the supplementary material anyway.

*4. Could you elaborate a bit more about the influence of horizontal resolution on the results?*

In addition to what is already discussed in the response to comment 1, we have added the maps showing the results at 100 km horizontal resolution to the supplementary material (Figs. S6-S9). The influence of the horizontal resolution to the results can be now better compared.

***Technical corrections***

*1. Line 13, abstract: delete 0.95*

The numerical values of the correlations in the abstract have been deleted.

*2. Line 11, page 7: delete repeated "the"*

Thanks for noticing this typo, the extra "the" has been deleted.

*3. Line 21 page 7. You could change "approximated by half-hour central differences" by "approximated by central differences" because your time step can change*

Yes, we agree with this. The "half-hour" has been deleted.

*4. Line 20 page 9: "with 25 km grid spacing" should be "with 25 km horizontal grid spacing"*

The word "horizontal" has been added in the place you mentioned.

[revised manuscript text omitted]